# Effects of Early Adverse Life Events on Depression and Cognitive Performance from the Perspective of the Heart-Brain Axis

**DOI:** 10.3390/brainsci13081174

**Published:** 2023-08-07

**Authors:** Yujie Xia, Gaohua Wang, Ling Xiao, Yiwei Du, Shanshan Lin, Cai Nan, Shenhong Weng

**Affiliations:** 1Department of Psychiatry, Renmin Hospital of Wuhan University, 238 Jiefang Rd., Wuhan 430060, China; xiaxy0517@163.com (Y.X.);; 2Institute of Neuropsychiatry, Renmin Hospital of Wuhan University, 238 Jiefang Rd., Wuhan 430060, China; 3Taikang Center for Life and Medical Sciences, Wuhan University, China Donghu Road No. 115, Wuhan 430071, China

**Keywords:** depression, cardiac autonomic regulation, early adverse life events, cognitive function, electroencephalograms, heart rate variability

## Abstract

Early adverse life events (EALs) increase susceptibility to depression and impair cognitive performance, but the physiological mechanisms are still unclear. The target of this article is to clarify the impact of adverse childhood experiences on emotional and cognitive performance from the perspective of the heart–brain axis. We used the Repeatable Battery for the Assessment of Neuropsychological Status (RBANS) to test cognitive function and the Childhood Trauma Questionnaire (CTQ) to assess adverse childhood experiences. Heart rate variability (HRV) and electroencephalograms (EEG) were acquired at rest. We observed that subjects with depression had experienced more traumatic events during their childhood. Furthermore, they exhibited lower heart rate variability and higher power in the delta, theta, and alpha frequency bands. Moreover, heart rate variability partially mediated the association between childhood trauma exposure and depressive symptoms. Our findings suggested that adverse life events in childhood could influence the development of depression in adulthood, which might be linked to cardiac autonomic dysfunction and altered brain function.

## 1. Introduction

Early adverse life events (EALs) are an external cause of susceptibility and play an important part on a variety of psychological and physical disorder occurrences [1]. Children who have experienced early trauma are at a higher risk of developing depression and frequently report impairments in cognitive abilities, such as attention, memory, and inhibition [2,3,4]. Nevertheless, there is still dispute about the magnitude of the impairment of emotional and cognitive function as a result of trauma. Whether there is a cumulative effect of trauma and depression on cognitive function remains uncertain. More importantly, the mechanisms through which EALs impact emotional and cognitive performance are currently unclear.

A mounting body of evidence suggests that there is an intimate crosstalk between the heart and the brain, which could be an underlying physiological mechanism of emotional and cognitive impairment caused by EALs [5]. Furthermore, this complex biological crosstalk may be reflected in changes in the cardiac autonomic nervous system. The cardiovascular autonomic nerve system serves as a bridge between heart and brain. The brain controls the heart directly through the sympathetic and parasympathetic branches of the autonomic nervous system.

Heart rate variability (HRV) is a non-invasive indicator of cardiac control by the autonomic nervous system [6]. It is defined as the naturally occurring fluctuation in the time intervals between successive heart beats over time. Time-domain, frequency-domain, and non-linear measures can all be used to estimate HRV [7]. According to the neurointernal integration model, HRV has been highlighted as a helpful biosignal indicator to reflect mental health status, reflecting the primary output of the central autonomic network [8,9]. Numerous studies have suggested that HRV is significantly reduced in patients with several psychiatric disorders, especially for major depression [10]. On the other hand, HRV is also suggested to reflect executive control functions such as inhibiting unwanted responses and updating [11]. Several behavioral task studies have established that higher resting HRV is associated with more adaptive executive control over stimuli. The central nervous system communicates with the autonomic nervous system through interoceptive neural circuits that contribute to physiological functions beyond homeostatic control, for a range of physiological functions, including emotional regulation and cognitive function [12,13].

Band power reflects the power or energy within specific frequency bands of the brain’s electrical activity. Analyzing band power helps understand the dominance or suppression of specific brain oscillations. It is measured directly using electroencephalography (EEG). Recent studies have discovered that each band is associated with distinct mechanisms. Delta powers indicate deep sleep. Theta powers reflect emotional processing. Alpha power is associated with inactivity and relaxation in various clinical conditions. Beta powers signify anxiety and introverted attention, while gamma powers are linked to attention and sensory systems, and can also reflect mood swings [14,15,16,17]. Research using EEG found that childhood trauma was significantly associated with left frontal alpha power and left parietal theta power at rest [18]. These studies illustrated a link between traumatic experiences and abnormalities in brain regions involved in autonomic control and stress response. Additionally, studies have shown a negative correlation between frontal EEG power density at beta frequencies and R-wave to pulse interval, indicating a relationship between cortical tone and cardiac influences [19]. Furthermore, another study found a negative correlation between Rolandic low-frequency beta rhythms and sympathetic activity [20].

Currently, it is uncertain how childhood trauma impacts affective and cognitive performance. Therefore, this study aims to assess the impact of childhood traumatic experiences on emotional and cognitive performance from the perspective of the heart–brain axis. Childhood trauma association with depression severity, cognitive performance, and heart rate variability indicators is hypothesized based on the existing literature. This change is believed to be related to the interaction between the brain and the heart.

## 2. Materials and Methods

### 2.1. Participants

The study was conducted at the Renmin Hospital of Wuhan University, Wuhan, China, between March and June 2021. Patients were recruited from the hospital’s psychiatric outpatient treatment unit if they met the following criteria: they were between 18 and 55 years old and experiencing a depressive episode. All patients were drug-naïve or had not taken medication regularly for more than two weeks. Exclusion criteria for depression patients included a lifetime history of bipolar affective disorder or psychotic disorder, an acute life-threatening suicidal crisis, a mental deficiency, and a severe organic disorder, or current pregnancy and breastfeeding. The diagnosis was determined using the Structured Clinical Interview for Axis I DSM-IV-TR Disorders (SCID)—Patient Edition and Hamilton Depression Scale score ≥seven at screening.

Healthy controls were recruited from university students and the general community through online advertisements if they met the following criteria: they were between 18 and 55 years old, had no history of any psychiatric disorders, and were not using psychotropic medication. Exclusion criteria for healthy controls included a history of any psychiatric disorder, use of psychotropic medication, a positive familial history (first-degree relatives) for psychotic disorder, a mental deficiency, and a severe organic disorder, or current pregnancy and breastfeeding. The diagnosis was determined using the Structured Clinical Interview for Axis I DSM-IV-TR Disorders (SCID)—Patient Edition and Hamilton Depression Scale score <seven at screening.

The total sample comprised 97 participants, divided into two distinct groups: the depression group (n = 48) and healthy controls (n = 49). Groups did not differ in age, gender, or years of education. The protocol and informed consent for this study was approved by Renmin Hospital of Wuhan University (approval No. WDRY2020-K236). All participants provided written informed consent and were matched with NIMH advocate from the Human Subjects Protection Unit to monitor consent and participation.

### 2.2. Psychological and Cognitive Function Measures

The study used the Hamilton Depression Scale (HAMD) to evaluate the severity of depressive symptoms using a 5-point scale from 0 to 4 [21]. A total score of ≥7 indicated depressive symptoms or a higher likelihood of depression. To ensure objectivity and consistency, two professional raters independently evaluated the scale through conversation and observation.

The Chinese-validated version of the Childhood Trauma Questionnaire (CTQ) was used to assess childhood trauma [22]. The CTQ measures emotional, physical, and sexual abuse, as well as emotional and physical neglect, with additional minimization/denial subscales that measure extreme response bias. It consists of 28 items and is rated on a 5-point Likert scale, with a possible score range of 28–140. Despite being assessed retrospectively, the CTQ is a well-established scale for evaluating the history of childhood trauma. To assess reliability, Cronbach’s Alpha was calculated, which in this study was 0.85 for internal consistency reliability.

Cognitive function was assessed by a trained psychiatrist using the Repeatable Battery for the Assessment of Neuropsychological Status (RBANS), which includes twelve sub-tests resulting in five index scores and a total score [23]. The test index scores measure immediate memory (story memory and listing learning tasks), visuospatial/constructional ability (line orientation and figure copy tasks), language (semantic fluency and picture naming tasks), attention (coding and digit span tasks), and delayed memory (list recall, story recall, figure recall, and list recognition tasks). To assess internal consistency, Cronbach’s Alpha was calculated, which was 0.76, indicating good internal consistency of the RBANS.

### 2.3. Heart Rate Variability

To assess HRV, electrocardiography was recorded in a quiet, temperature-controlled room. A Heart Rate Variability Analysis System (Media 300, Guangzhou, China) was used to acquire the records. Before the test, the subjects were advised to avoid drinking tea and coffee and to avoid intense exercise. The HRV was measured for a short period after 15 min of rest at rest. The following variables were extracted.

Time-domain: (a) root mean square of the successive differences (RMSSD), which reflects higher variability and cardiac parasympathetic activity; (b) standard deviation of the RR intervals (SDNN, in milliseconds), which reflects the combined sympathetic and vagal activity. Frequency domain; (c) low-frequency (LF, 0.04–0.15 Hz) represents an amalgam of sympathetic and parasympathetic activity; (d) high-frequency (HF, 0.15–0.40 Hz) represents parasympathetic activity; (e) the LF/HF ratio represents the sympathovagal balance. Poincaré plot: the Poincaré plot is a scatterplot in which each N–N interval is plotted as a function of the previous one, which provides both a qualitative and quantitative analysis of HRV. The shape of the plot can be used to quantify the parameters SD1 and SD2; (f) SD1 represents the dispersion of the points perpendicular to the line of identity, and it is thought to be an index of the instantaneous beat-to-beat variability; (g) SD2 represents the dispersion of the points along the line of identity, and it represents the slow variability of heart rate [24].

### 2.4. EEG Recording and Preprocessing

Resting-state EEGs were acquired in a quiet, electromagnetically shielded room. Participants were required to be awake and to avoid movement during EEG acquisition. Recordings were made by 64 channels of active electrodes at a sampling rate of 1000 Hz, amplified from DC to 100 Hz using ANDETE (Millionandey, Guangzhou, China) amplified. The electrodes were referenced to the average of the bilateral mastoid during acquisition and re-referenced offline to the average of the whole brain. Statistical analysis of the data was based on the following electrodes: frontal (F3, Fz, F4, FC3, FCz, and FC4), central (C3, Cz, C4, CP3, CPz, and CP4), parietal (P3, Pz, P4, PO3, PO4, and POz), and occipital (O1, Oz, and O2). All of the electrode impedances were below 5 kΩ. The EEG data were visually inspected, and segments containing artifacts were excluded. The EEG data were then preprocessed using a 0.1 to 70 Hz bandpass filter to improve the signal-to-noise ratio. Corrections for ocular artifacts were made using the Independent Component Analysis (ICA) procedure.

In power spectra analysis, five types of frequency bands are defined: delta (from 1 to 4 Hz), theta (from 4 to 8 Hz), alpha (from 8 to 13 Hz), beta (from 13 to 30 Hz), low-gamma (from 30 to 52 Hz), and high-gamma (from 52 to 70 Hz). For each subject, four components were separated using Fast Fourier Transform (FFT), and the absolute powers were averaged over all time windows and frequency bands. To investigate the relationship between abnormal EEG rhythm oscillation and depressed mood, a cluster-based permutation test was conducted to compare the spectral power of 61-channel EEGs between MDD and HC subjects. Data correction was done by cluster, and a permutation test was conducted on each electrode. Pearson’s test was used to determine the association between childhood trauma and EEG rhythm oscillation (*p* < 0.05), The False Discovery Rate (FDR) was used in multiple comparisons to control the rate of Type I errors in null hypothesis testing. EEG data were preprocessed and analyzed offline using MATLAB 2020b and the EEGLAB toolbox.

### 2.5. Data Analysis

The social demographics of depressed and healthy participants were compared using the chi-square test for categorical variables and analysis of variance (ANOVA) for continuous variables. Continuous data were represented using mean and standard deviation (Mean ± SD), while non-normally distributed variables were described using medians and percentiles. Internal consistency of the RBANS was assessed using Cronbach’s Alpha for reliability analysis, and the CTQ between the two groups was compared using ANOVA, with group (MDD versus HC) as the group variable and score on a scale as a dependent variable. To investigate the association of childhood trauma with cognitive function and psychophysiological measures, with gender as a control factor, partial correlations were used. To investigate the complex relationships between variables, mediation analysis for the overall sample was used. By assessing these associations, we could identify direct and indirect effects and explore potential mediating mechanisms. The following associations were included: association between EALs and LF-HRV (path a); association between LF-HRV and depressive symptoms (path b); association between EALs and depressive symptoms (path c); indirect effect of EALs on depressive symptoms through LF-HRV (path ab). Standardized regression coefficients (β), standard error (SE), and confidence intervals (CI) were reported to establish the statistical significance of the relationships. The significance level was set at *p* < 0.05 for all analyses, and IBM SPSS Statistics 26.0 for Windows was used for two-tailed analysis with a significance level of *p* < 0.05; partial η2 or Cohen’s d2 were reported for significant effects.

## 3. Results

### 3.1. Demographic and Clinical Characteristics of the Two Groups

Table 1 shows the social-demographic data for MDD and HC; there were no statistically significant differences between the two groups on any demographic variable. Additionally, Table 1 displays the mean ± SD of CTQ scores, which were significantly higher for MDD than for HC. We conducted univariate ANOVA with a group (MDD versus HC) as the group variable and cognitive measure as the dependent variable.

The mean ± SD of RBANS scores is shown in Table 2. MDD showed significantly lower scores than HC on the RBANS total score and subscale scores on immediate memory, language, and attention, except for visuospatial/constructional and delayed memory scores.

Table 3 shows the frequency domain, time domain, and non-linear indexes of the two groups, with healthy controls showing significantly higher scores than the depression group. Specifically, significant differences were found between the two groups in the following indexes: SDNN, RMSSD, LF, HF, SD1, and SD2.

### 3.2. Analysis of the Correlation between EALs and Clinical Characteristics

As illustrated in Figure 1, we found increased delta, theta, alpha, and beta power of the EEG in MDD subjects. Low gamma power of the EEG was significantly reduced in MDD subjects. Specifically, delta power was significantly abnormal in almost every brain region of MDD subjects compared with HC subjects. Theta band abnormalities were mainly observed in the frontal and central areas, while alpha band abnormalities were mainly observed in the frontal areas (FC3, Fz, FCz, and C3). Beta band abnormalities were mainly observed in the central and parietal areas (FC3, C3, Pz, and P4). Low-gamma power was significantly reduced in the central areas (Cz). No significant difference was found in high-gamma power (*p* > 0.05).

We found that the total CTQ score was significantly correlated with visuospatial/constructional (r = −0.261, *p* = 0.010), language (r = −0.326, *p* = 0.001), attention (r = −0.202, *p* = 0.049), and RBANS total score (r = −0.390, *p* = 0.000). In addition, the total CTQ score was significantly correlated with LF (r = −0.241, *p* = 0.018), HF (r = −0.223, *p* = 0.028), SDNN (r = −0.222, *p* = 0.029), and RMSSD (r = −0.208, *p* = 0.041). There were no significant interactions or main effects of age or fitness on other HRV indices. Figure 2 shows the correlation between childhood trauma and EEG power spectra during rest conditions in subjects with depressive disorder. The following correlations were observed: alpha absolute power during rest condition was significantly positively correlated with childhood trauma in certain brain regions (0.310 to 0.400; FC1, FPz, and PZ); beta and delta absolute power during rest condition were positively correlated with childhood trauma in several electrodes (0.302 to 0.420; CZ); low-gamma and high-gamma absolute power during rest condition were positively correlated with childhood trauma in several electrodes (0.302 to 0.420; FC1 and CZ); theta absolute power during rest condition was not significantly positively correlated with childhood trauma.

### 3.3. Mediation Analyses

As illustrated in Figure 3, the mediation model for the entire sample was used to test the central hypothesis of the study. The purpose is to explore the relationships in a comprehensive manner. EALs had a statistically significant effect on depressive symptoms (path c in Figure 1; β = 0.514, SE = 0.085, *p* = 0.000, 95% CI [0.282, 0.618]). Childhood trauma exposure was associated with more significant depressive symptoms. EALs also had a statistically significant effect on LF-HRV (path a; β = −0.235, SE = 2.961, *p* = 0.021 95% CI [−12.73, −1.12]), indicating that childhood trauma exposure predicted lower LF-HRV. Additionally, the effect of HRV on depressive symptoms was also statistically significant, whereby lower LF-HRV was associated with greater depressive symptoms (path b; β = −0.20, SE = 0.003, *p* = 0.027, 95% CI [−0.012, −0.001]). After controlling for HRV, the direct effect of childhood trauma exposure on depressive symptoms was reduced compared to the total effect (path c’; β = 0.467, SE = 0.086, *p* = 0.000, 95% CI [0.282, 0.618]). The indirect effect of childhood trauma exposure on depressive symptoms was statistically significant (path ab; β = 0.045, SE = 0.028, *p* = 0.108, 95% CI [0.002, 0.110]). Taken together, these results suggest that HRV partially mediated the association between childhood trauma exposure and the frequency of depressive symptoms. This model accounted for 28.7% of the variance in depressive symptoms.

## 4. Discussion

This study aims to assess the impact of childhood traumatic experiences on the emotional and cognitive performance of individuals with depression in a dual physical-psychometric dimension. We found that EALs have a significant influence on the development of depression in adulthood. This effect may be associated with cardiac autonomic dysfunction and altered brain function.

### 4.1. Effect of EALs on Depression

EALs can predict the severity of depression symptoms and decrease the threshold for depressive reactions to stressors, which aligns with previous literature [25]. Studies have also found that EALs significantly influence the risk of adult psychopathology [26]. However, it is currently unclear whether depression is associated with specific stressors. We have found that depression severity appears to be more associated with emotional neglect. Similarly, some studies associate MDD with exposure to certain types of childhood trauma, including sexual abuse, emotional abuse, and family conflict [27,28]. Not all studies support this conclusion; some studies concluded that mental illnesses were not related to specific childhood adversities [29]. Rather, it may depend on stressor severity, chronicity, and exposure age. Only stressors that result in significant or sustained stress responses, characterized by the HPA axis or other immune system activity, would influence vulnerability to psychiatric illnesses.

### 4.2. Effect of EALs on Cognitive Function

Comparison of cognitive function between depressed patients and healthy controls showed significant differences in global cognitive function, with the exception of visuospatial/constructional and delayed memory scores. This finding is consistent with earlier findings that the total RBANS score was significantly lower in the MDD group than in the HC group [30]. Furthermore, several meta-analysis have indicated that cognitive deficits were significantly associated with depressive severity [31]. In addition, we found that there were significant differences in immediate memory between the two groups; no group difference was found in delayed memory. When interpreting this divergence, several characteristics should be considered. Firstly, immediate memory is associated with attention, with depressed patients reflecting attentional arousal and sustained attention rather than memory impairment. Similarly, some studies have suggested that impairments in memory seem unrelated to depression severity [32]. There is debate about whether it could be a trait marker or a risk factor for depression. Furthermore, the relatively young age of the present sample (mean age, 21.36 years) may also have been an underlying factor. Available data have claimed that differences in cognitive performance between depression patients and healthy individuals are more pronounced in older samples than in younger samples [33].

Finally, we found that EALs were negatively associated with global cognitive function, but not with memory domain. A recent meta-analysis reported that the timing of trauma was a significant factor in cognitive impairment, suggesting that EALs can impair brain function, particularly attention and visuospatial function [32,34]. A functional magnetic resonance imaging study showed that EALs resulted in aberrant function of the visual cortical areas involved in attention, suggesting that disturbances in specific brain networks underlie cognitive abilities [35].

### 4.3. Effect of EALs on the Heart–Brain Axis 

The present research reports reduced HRV indexes, derived from both the time and frequency domains of depressed patients. We found resting state vagal activity to be more strongly reduced in individuals exposed to childhood trauma, which is consistent with our previous hypothesis that a dysregulation of the autonomic nervous system is associated with EALs. Most importantly, our research has shown that HRV partially mediated the association between EALs and the frequency of depressive symptoms. Specifically, childhood trauma exposure was associated with HRV, which predicted more significant depressive symptoms. LF partially mediated the association between EALs and the frequency of depressive symptoms. Previous studies have suggested that LF can reflect a complex combination of sympathetic, parasympathetic, and other unidentified factors [36], and as only LF is involved, not HF and not RMSSD, it can be assumed that sympathetic nerves play a mediation effect.

Traditional theories have proposed a close relationship between HRV and emotion regulation [8]. On a central nervous level, the overlap between the network’s mediating autonomic and affective regulation illustrates this interaction. Particularly, the ventromedial prefrontal cortex, insula, amygdala, and cingulate are responsible for the autonomic regulation of heart rate and emotional processing. Individuals with higher HRV have improved emotion-regulating ability [37]. This is supported, for example, by the findings of an association between higher resting RSA and greater self-reported emotion regulation, as well as the use of more constructive and less passive coping strategies [38]. Individuals with higher HRV displayed lower traits and social anxiety [13]. In contrast, low HRV was related to an enhanced startle response, indicating exaggerated anticipatory anxiety. Previous studies have identified EALs as a risk factor for abnormal emotion regulation, with HRV serving as an indicator of abnormal emotion regulation. Hence, we hypothesize that EALs contribute to depressive symptoms through maladaptive emotional processing.

Furthermore, we found that patients with depression exhibited more neurodevelopmentally immature spectral power profile patterns, characterized by a relatively higher spectral power in lower frequency bands (delta (from 1 to 4 Hz), theta (from 4 to 8 Hz), alpha (from 8 to 13 Hz)), and relatively lower spectral power in a higher frequency band (low-gamma (from 30 to 52 Hz), high-gamma (from 52 to 70 Hz)), and the CTQ scores showed significant positive correlations with alpha powers in the anterior regions [18]. A recent study also reported that depressed patients had higher alpha values on the left side of the brain compared to healthy controls, which is consistent with Hosseinifard’s discovery that alpha and theta bands can be used to distinguish depressed patients from healthy controls. These bands are thought to be associated with emotional processing [39].

However, clinical applications of EEG in patients with childhood traumatic experiences remains a matter of debate. It has been suggested that physical and emotional neglect during childhood is associated with increased cortical arousal in adulthood and decreased alpha power in the left parietal cortex. However, prospective research showed relatively greater alpha-band spectral power in the group with higher-risk parenting and family adversity. The heterogeneous characteristics of the target group will be an important confounding factor in this discrepancy. In the future, better-designed studies controlling for factors such as when childhood trauma occurs, its duration, type, and severity will be required to solve this discrepancy.

The improvement of heart rate variability (HRV) to alleviate depressive symptoms has been the primary focus of clinical research. It has been suggested that training in the alpha band through neurofeedback can enhance HRV and improve working memory [40]. Additionally, engaging in physical activity has been found to be a significant protective factor that enhances mood and cognitive function [41]. Mindfulness meditation and biofeedback therapy have also demonstrated the ability to increase HRV and alleviate depressive symptoms [42,43].

Different pathways might be involved, considering potential patterns of association between EALs and psychopathology. It is possible to assume that EALs cause insufficient recruitment of the prefrontal cortex (PFC) and poor emotion control, which result in psychopathology and low HRV. However, we could also hypothesize that early traumatic experiences can result in aberrations in the development of the ANS, specifically, an insufficient rise in vagal activity during childhood and adolescence, which in turn results in inadequate PFC maturation and deficiencies in affective regulation, ultimately raising the risk for psychopathology. Future longitudinal studies addressing childhood trauma, HRV, and brain development are necessary to clarify the same cascade because our study was unable to do so.

### 4.4. Limitations

The study utilized a retrospective, self-reported CTQ scale that failed to account for specific stressful episodes or quantify intensity or chronicity. This reliance on self-reporting can be influenced by memory bias, which can be severe in depressed individuals. Furthermore, due to resource constraints, we were only able to recruit a small number of participants. As a result, the impact of mixed factors such as gender and age cannot be thoroughly discussed. Caution should be exercised in interpreting the findings due to the small sample size, which may limit the generalizability of the results. To investigate the impact of childhood trauma, a more effective approach would be to divide the depressed patients into subgroups based on traumatic experiences. Due to the limitation of the sample size, we could not explore this question thoroughly. In the future, we plan to increase the sample size to provide a more comprehensive answer. Finally, since our cross-sectional study only examined participants at one point in time, we were unable to assess the long-term developmental consequences of childhood trauma exposure on HRV or cognition. Longitudinal studies are needed to fully understand the effects of childhood trauma on mental and physical health over time.

## 5. Conclusions

We did find a complex relationship between EALs, cognitive function, and depression, which differed between diagnostic groups. However, these findings need to be replicated in independent populations to confirm their validity and generalizability. Future longitudinal studies should incorporate the physiological measures of the stress response, investigating which factors contribute to emotional regulation in EALs, and clarifying the relationship with the heart–brain axis. Finally, future research should examine the influence of emotional processes on cognitive performance and how early life adversities affect these relationships.

## Figures and Tables

**Figure 1 brainsci-13-01174-f001:**
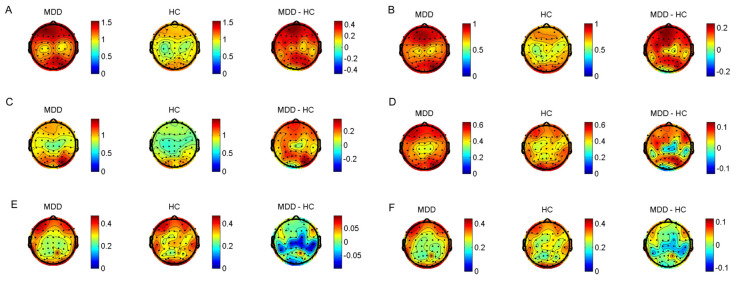
The difference between patients and controls in absolute powers. (**A**) Delta powers between MDD and controls. (**B**) Theta powers. (**C**) Alpha powers. (**D**) Beta powers. (**E**) Low−gamma powers. (**F**) High−gamma powers. MDD: Mean Power between subjects in the MDD group. HC: mean power between subjects in the HC group. MDD−HC: mean Power difference between subjects in the MDD minus HC group.

**Figure 2 brainsci-13-01174-f002:**
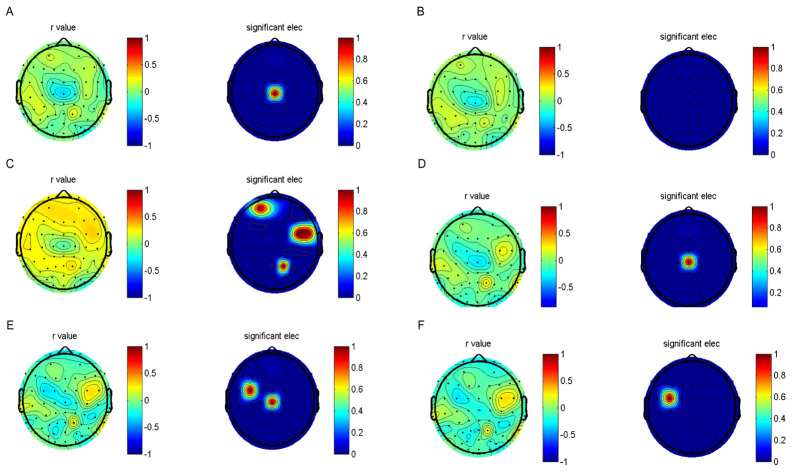
The results of the correlation analyses and electroencephalogram. The EEG topographic map after false discovery rete (FDR) correction. (**A**) Delta powers between MDD and controls. (**B**) Theta powers. (**C**) Alpha powers. (**D**) Beta powers. (**E**) Low−gramma powers. (**F**) High−gramma powers.

**Figure 3 brainsci-13-01174-f003:**
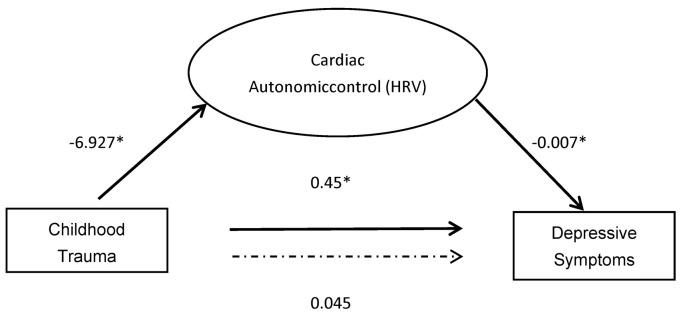
Mediation between childhood trauma and depressive symptoms through cardiac autonomic control. * significant (*p* < 0.05).

**Table 1 brainsci-13-01174-t001:** Social-demographics of participants.

Variables	Depressed Subjects (n = 48)	Healthy Controls (n = 49)	F or X 2	*p*
Gender (male/female)	15/33	23/26	1.768	0.184
Age (years)	21.06 ± 2.60	21.53 ± 2.18	0.432	0.513
Education (years)	14.69 ± 1.80	15.27 ± 1.55	2.458	0.120
BMI (kg/m^2^)	21.36 ± 3.43	20.51 ± 2.59	1.918	0.169
Drinking (drinker/nondrinker)	5/44	3/45	0.501	0.479
Smoking (smoker/nonsmoker)	3/45	4/45	0.133	0.716
emotional abuse	10.09 ± 4.31	6.45 ± 1.99	6.54	0.000
physical abuse	7.58 ± 3.57	5.80 ± 2.08	2.76	0.003
sexual abuse	6.52 ± 2.32	5.43 ± 1.15	2.00	0.004
emotional neglect	14.13 ± 4.63	9.22 ± 4.87	5.07	0.000
physical neglect	9.50 ± 3.93	6.82 ± 2.65	3.94	0.000
CTQ sum score	48.40 ± 13.20	33.96 ± 9.36	6.23	0.000

**Table 2 brainsci-13-01174-t002:** Descriptive values of cognitive performance and group differences.

RBANS Score	Depressed Subjects (n = 48)	Healthy Controls (n = 49)	F	*p*
Immediate memory	86.95 ± 10.98	92.80 ± 10.73	6.99	0.010
Visuospatial/constructional	93.98 ± 15.53	98.78 ± 13.11	2.68	0.105
Language	89.32 ± 14.07	100.29 ± 13.10	15.62	0.000
Attention	110.60 ± 15.94	117.27 ± 10.85	5.78	0.018
Delayed memory	89.87 ± 14.14	94.35 ± 10.79	3.05	0.084
RBANS total score	91.68 ± 9.87	100.43 ± 9.97	18.63	0.000

**Table 3 brainsci-13-01174-t003:** Descriptive values of heart rate variability and group differences.

	Depressed Subjects (n = 48)	Healthy Controls (n = 49)	t	*p*
SDNN (ms)	40.00 (33.00, 52.00)	49.00 (40.00, 63.00)	2.72	0.003
RMSSD (ms)	28.00 (19.00, 39.00)	33.00 (27.00, 45.00)	2.10	0.026
LF (ms^2^)	302.40 (171.91, 562.27)	581.62 (396.71, 834.34)	3.85	0.000
HF (ms^2^)	230.93 (129.32, 552.77)	301.95 (243.78, 630.03)	2.02	0.038
LF/HF	2.40 (1.55, 3.25)	3.47 (1.59, 5.34)	1.07	0.30
SD1	19.79 (13.52, 27.35)	24.35 (19.16, 33.45)	2.20	0.028
SD2	50.49 (41.43, 70.01)	66.48 (54.04, 84.18)	3.09	0.002

## Data Availability

The data presented in this study are available on reasonable request from the corresponding author. The data are not publicly available due to a lack of patients’ consent to public data sharing.

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
