# Peer review of "Effects of Early Adverse Life Events on Depression and Cognitive Performance from the Perspective of the Heart-Brain Axis"

_brainsci, 2023, doi:10.3390/brainsci13081174_

Round 1

Reviewer 1 Report

Dear authors,

Congratulations on your work. I hope to help improve your article with the following suggestions.

Introduction

Honestly, the way you introduce HRV and band power could be more precise. Please, try to rewrite it so it can be read more pleasingly.

Once you talk about the bands, it would be better to do it by order and not randomly.

Materials and Methods

You need to explain what the inclusion criteria are.

EEG

I do not understand why you used a 0.1 to 30 Hz bandpass filter when you recorded the gamma band also…

Moreover, there is a gap between theta and alpha bands of 1 Hz. That 1 Hz can be more than enough to change some results of your study. Please, correct it.

Results

In line 187, you should have included a comma.

It is optional, but it would be better to intercalate the text about the tables and figures with the table and figures. For example: Text about Table 1 and after that Table 1. Text about Table 2 and after that Table 2. Etc.

Remember to add units.

Again, in Figure 1, it would be better if the bands were presented in their natural order and not randomly (the same as in Figure 2).

In Figure 2, you mention the FDR correction, which should be mentioned in the statistical analysis. Please correct it.

Discussion

Lines 278 to 280. The sentence needs to be more sound. Please, rewrite.

Line 323. You have lost a comma.

Finally, I would like to suggest you write a small paragraph on how we can improve HRV (since it seems to be somehow related to depression to a certain degree). For example, in your last paragraphs, before limitations, you already introduced that “clinical applications of EEG in patients with childhood traumatic experiences remain a matter of debate”. I recall at least, 2 promising ways to improve HRV, which in a certain way could have an impact on depression. For instance, you have neurofeedback training, which was revealed to enhance HRV when training in the alpha band and, at the same time, improved working memory (1 Domingos et al., 2021) and physical exercise as a protector variable (2 Santos et al., 2022).

1 Domingos, C., Silva, C. M. D., Antunes, A., Prazeres, P., Esteves, I., & Rosa, A. C. (2021). The influence of an alpha band neurofeedback training in heart rate variability in athletes. International Journal of Environmental Research and Public Health, 18(23), 12579.

2 Santos, J., Ihle, A., Peralta, M., Domingos, C., Gouveia, É. R., Ferrari, G., ... & Marques, A. (2022). Associations of physical activity and television viewing with depressive symptoms of the European adults. Frontiers In Public Health, 9, 799870.

Author Response

Response to Reviewer 1 Comments

Point 1: the way you introduce HRV and band power could be more precise. Please, try to rewrite it so it can be read more pleasingly.

Response 1: This section has been revised based on your suggestions, which can be found in lines 47 -77.

Point 2: Once you talk about the bands, it would be better to do it by order and not randomly.

Response 2: We have sorted them by bands as you suggested to be more organized. 

Point 3: You need to explain what the inclusion criteria are.

Response 3: The inclusion criteria have been outlined in the methods section of the article(in line 92) with the following inclusion criteria being described in detail.

  • Diagnosis of depression: This is typically a fundamental criterion for inclusion in a study related to depression. The diagnosis should be made by a qualified healthcare professional and be based on recognized diagnostic criteria,such as the Structured Clinical Interview for Axis I DSM-IV-TR Disorders (SCID)–Patient Edition.
  • Age: between the ages of 18 and 55.
  • Severity of depression: The severity of the depression can also be a criterion. This is assessed using a validated depression rating scale, such as the Hamilton Depression Scale (HAMD) .
  • Ability to give informed consent: Participants must be able to understand the study and its potential risks and benefits, and to make an informed decision about whether to participate .
  • Other health conditions: Depending on the study, patients with certain physical or mental health conditions may be excluded.

Point 4: I do not understand why you used a 0.1 to 30 Hz bandpass filter when you recorded the gamma band also… Moreover, there is a gap between theta and alpha bands of 1 Hz. That 1 Hz can be more than enough to change some results of your study. Please, correct it.

Response 4: We sincerely apologize for the error in the 0.1 to 30 Hz bandpass filter and the frequencies of theta (4–7 Hz) that were incorrectly written in the paper. We acknowledge that this was due to our carelessness during the writing process. In fact, we used 0.1-70 Hz bandpass filter and the frequencies of theta (4–8 Hz).Despite the mistake mentioned, we have verified that the data analysis and results remain valid and unaffected.

We understand the importance of accurately reporting the details in our research, and we deeply regret any confusion caused by this oversight. In light of your comments, we have carefully revised the relevant sections of the paper to rectify this error and ensure the accuracy of the information presented.

Point 5: In line 187, you should have included a comma.

Response 5: Thank you for your rigor, revisions have been done.

Point 6: It is optional, but it would be better to intercalate the text about the tables and figures with the table and figures. For example: Text about Table 1 and after that Table 1. Text about Table 2 and after that Table 2. Etc.

Response 6: Thank you for your advise, revisions have been done.

Point 7: Remember to add units.

Response 7: Thank you for your rigor, revisions have been done.

Point 8: Again, in Figure 1, it would be better if the bands were presented in their natural order and not randomly (the same as in Figure 2).

Response 8: I have sorted them by bands as you suggested to be more organized. 

Point 9In Figure 2, you mention the FDR correction, which should be mentioned in the statistical analysis. Please correct it.

Response 9 : The following revisions were made :

“The False Discovery Rate (FDR) was used in multiple comparisons to control the rate of Type I errors in null hypothesis testing. ”(in line 168)

Point 10: Lines 278 to 280. The sentence needs to be more sound. Please, rewrite.

Response 10 : The following revisions were made :

EALs can predict the severity of depressive symptoms and decrease the threshold for depressive reactions to stressors, which aligns with previous literature.

Point 11: Line 323. You have lost a comma.

Response 11 :Thank you for your work, revisions have been done.

Point 12:Finally, I would like to suggest you write a small paragraph on how we can improve HRV (since it seems to be somehow related to depression to a certain degree). For example, in your last paragraphs, before limitations, you already introduced that “clinical applications of EEG in patients with childhood traumatic experiences remain a matter of debate”. I recall at least, 2 promising ways to improve HRV, which in a certain way could have an impact on depression. For instance, you have neurofeedback training, which was revealed to enhance HRV when training in the alpha band and, at the same time, improved working memory (1 Domingos et al., 2021) and physical exercise as a protector variable (2 Santos et al., 2022).

1 Domingos, C., Silva, C. M. D., Antunes, A., Prazeres, P., Esteves, I., & Rosa, A. C. (2021). The influence of an alpha band neurofeedback training in heart rate variability in athletes. International Journal of Environmental Research and Public Health, 18(23), 12579.

2 Santos, J., Ihle, A., Peralta, M., Domingos, C., Gouveia, É. R., Ferrari, G., ... & Marques, A. (2022). Associations of physical activity and television viewing with depressive symptoms of the European adults. Frontiers In Public Health, 9, 799870.

Response 12 : The following revisions were made :

The improvement of heart rate variability (HRV) to alleviate depressive symptoms has been the primary focus of clinical research. It has been suggested that training in the alpha band through neurofeedback can enhance HRV and improve working memory. Additionally, engaging in physical activity has been found to be a significant protective factor that enhances mood and cognitive function. Mindfulness meditation and biofeedback therapy have also demonstrated the ability to increase HRV and alleviate depressive symptoms.(in line 362-368)

Reviewer 2 Report

Thank you very much for the opportunity to review this paper which attempts to shed light on possible neurobiological pathways linking early childhood adverse experiences and later life depression.

Early Adverse Life Events (EALS) is a less usual abbreviation that Adverse Childhood Experiences (ACES)- can the authors explain why they use this term?

Introduction Paragraph 2 there are  minor English grammar  errors- could you change to “…suggests that there is an intimate crosstalk…”

“Furthermore, this complex biological crosstalk may be reflected in changes in the cardiac autonomic nervous system.”

Para 3 “band power is one direct piece of information...”

Can the authors explain for readers not familiar with the term what exactly is meant by ‘band power?’ 

Materials and Methods:

The authors state that patients were recruited through the hospital’s psychiatric outpatient treatment unit, then later state that the sample was divided into the Depression group and the healthy controls.  I am unsure where the healthy controls were recruited from and how?  if they were recruited from outpatients, they were likely not healthy controls.  Can the authors clarify?

This aspect of the design- where depressed patients are compared with healthy controls puzzled me.  If the main research question is- does a history of ACEs result in changes in the heart brain axis that could be somehow involved in the pathway to depression and cognitive impacts, then a better design may be to compare (a) depressed patients with history of childhood trauma to depressed patients with no history of any trauma, and/ or (b) an unselected group of subjects who have a history of childhood trauma compared with an unselected group with no history of childhood trauma and to compare the incidence of depression, and whether there are differences between the two groups in terms of heart rate variability and EEG patterns? Can the authors clarify?

Results
In the depressed group, there are 15 males and 33 females, while in the control group there are 23 males and 26 females yet surprisingly this difference is not statistically significant.  Could the authors confirm and comment- could this difference ,even if not achieving statistical significance still impact the results?

Tables 1,2 and 3 compare depressed subjects with controls, but it is unclear how any of these parameters relate to early life adversity.

The mediation analysis attempts to address this- but was this conducted only on the depressed patients?  Were these same parameters looked at for the healthy controls?  Did controls with the same adversity exposures differ from depressed subjects in any of these metrics?

There are a very large number of non-standard abbreviations used throughout the paper, requiring a lot of back and forth to follow the text.  I would suggest a prominent box at the start of the manuscript that lists these abbreviations and can be referred back to by the reader.  If that is not possible, the full descriptions of the variables will need to be given in the Results and discussion sections.

Although the findings are certainly interesting, it was unclear how many associations were tested to identify the significant ones- could the authors clarify?

The last line of the Conclusions section is part of the manuscript instructions and should be deleted.

Some of the conclusions being drawn in the paper are based on very small numbers of subjects- this should be stated.

Informed Consent statement: This does not appear to be correct- it repeats the statement on data availability.  Can the authors revise?

However, the approach is somewhat novel and the findings certainly interesting and clinically relevant.  As the authors state, longitudinal studies are needed to confirm these observations on larger groups.

Minor edits suggested.

Author Response

Response to Reviewer 2 Comments

Point 1: Early Adverse Life Events (EALS) is a less usual abbreviation that Adverse Childhood Experiences (ACES)- can the authors explain why they use this term

Response 1:The terms "Early Adverse Life Events" and "Adverse Childhood Experiences" both refer to negative events or experiences that occur during childhood and can have long-lasting effects on an individual's health and well-being. While they are related, there are some differences between the two:

Adverse Childhood Experiences (ACEs): ACEs are potentially traumatic events that occur during childhood. These experiences can include violence, abuse (emotional, physical, or sexual), neglect, household dysfunction (such as growing up with substance abuse, mental illness, or incarcerated household members), or other forms of adversity1.

Early Adverse Life Events(EALs): The term "Early Adverse Life Events" is not as widely used or defined as ACEs. However, it generally refers to negative experiences or events that occur early in life, including during childhood. These events can encompass a broader range of adverse experiences beyond the specific categories of ACEs, such as economic hardship, exposure to community violence, natural disasters, or other traumatic events that may not fit within the ACEs framework. The impact of early adverse life events on long-term health and well-being may vary depending on the specific nature of the event and the individual's resilience and support systems2 .

In summary, the results of this study cannot prove that depressive symptoms are associated with specific stressors, EALs were used to refer to nonspecific stress in childhood.

1.Narayan, A. J., Lieberman, A. F., & Masten, A. S. (2021). Intergenerational transmission and prevention of adverse childhood experiences (ACEs). Clinical psychology review, 85, 101997. https://doi.org/10.1016/j.cpr.2021.101997

2.Vergne, D. E., & Nemeroff, C. B. (2006). The interaction of serotonin transporter gene polymorphisms and early adverse life events on vulnerability for major depression. Current psychiatry reports, 8(6), 452–457. https://doi.org/10.1007/s11920-006-0050-y

Point 2: Introduction Paragraph 2 there are minor English grammar  errors- could you change to “…suggests that there is an intimate crosstalk…” “Furthermore, this complex biological crosstalk may be reflected in changes in the cardiac autonomic nervous system.”

Response 2:Thank you greatly for your job, I already made the corrections based on your suggestions.

Point 3: Para 3 “band power is one direct piece of information...”Can the authors explain for readers not familiar with the term what exactly is meant by ‘band power?

Response 3: Thank you greatly for your job,the following revisions were made :

Band power reflects the power or energy within specific frequency bands of the brain's electrical activity. Analyzing band power helps understand the dominance or suppression of specific brain oscillations. It is measured directly using electroencephalography (EEG).

Point 4: The authors state that patients were recruited through the hospital’s psychiatric outpatient treatment unit, then later state that the sample was divided into the Depression group and the healthy controls.  I am unsure where the healthy controls were recruited from and how?  if they were recruited from outpatients, they were likely not healthy controls.  Can the authors clarify?

Response 4: For reasons of the length of this article, the inclusion and exclusion criteria for healthy controls were not previously described in detail, and the following is a supplement of that section:

  • An equal number of healthy control subjects were recruited from the college or community via online advertisement.
  • Age: between the ages of 18 and 55.
  • Health conditions:neither had a history of any psychiatric disorder nor currently meet criteria for any severe organic disorder.
  • Ability to give informed consent: Participants must be able to understand the study and its potential risks and benefits, and to make an informed decision about whether to participate . 

Point 5:This aspect of the design- where depressed patients are compared with healthy controls puzzled me.  If the main research question is- does a history of ACEs result in changes in the heart brain axis that could be somehow involved in the pathway to depression and cognitive impacts, then a better design may be to compare (a) depressed patients with history of childhood trauma to depressed patients with no history of any trauma, and/ or (b) an unselected group of subjects who have a history of childhood trauma compared with an unselected group with no history of childhood trauma and to compare the incidence of depression, and whether there are differences between the two groups in terms of heart rate variability and EEG patterns? Can the authors clarify?

Response 5:According to the data we collected, the percentage of subjects without a history of childhood trauma was low in the depression group, and similarly the percentage of subjects with childhood trauma in the healthy groups was also low. Currently, the small sample size of our study data does not allow us to divide it into four groups to compare for subsequent analysis. This is one of the limitations of our study, however, the focus in our study was on the risk factors of depressed patients, therefore the subjects were first divided into two groups according to their disease and then the related factors were analyzed.

Point 6: In the depressed group, there are 15 males and 33 females, while in the control group there are 23 males and 26 females yet surprisingly this difference is not statistically significant.  Could the authors confirm and comment- could this difference ,even if not achieving statistical significance still impact the results.

Response 6: We appreciate your attention to detail and your insightful comment regarding the difference in gender distribution between the depressed and control groups.In our study, we observed that there were 15 males and 33 females in the depressed group, while the control group consisted of 23 males and 26 females. We acknowledge that this difference in gender distribution did not achieve statistical significance.

However, we agree with your question regarding the potential impact of this difference, even if it is not statistically significant. While statistical significance is an important measure in determining the likelihood of an observed difference being due to chance, it does not provide a complete picture of the practical or clinical relevance of the findings.

In our analysis, we mainly considered other factors such as HAMD scores, CTQ scores, RBANS scores, and measures related to heart rate variability, to compare the depressed and healthy groups. We found significant differences between the two groups in these measures.Though gender is still important to consider its potential impact on the results. Gender can be a significant factor in various psychological and physiological processes, and even small differences in gender distribution may have implications for the interpretation of our results.

We believe that although the difference in gender distribution did not reach statistical significance based on our findings. We will conduct a more detailed analysis of gender differences by increasing the sample size in future studies. If conditions permit, we plan to address this limitation and explore the potential impact of gender differences in a larger sample. This will provide a more comprehensive understanding of the role of gender in our findings.

Point 7: Tables 1,2 and 3 compare depressed subjects with controls, but it is unclear how any of these parameters relate to early life adversity.

Response 7 : As clinical doctors we are more concerned with the difference between the disease state and the healthy state. Next is the correlation between these factors.

This study was analyzed in the same way, firstly to find the differences between the two groups , and secondly to explain the relationship between these parameters and early life adversity.

Point 8:The mediation analysis attempts to address this- but was this conducted only on the depressed patients?  Were these same parameters looked at for the healthy controls?  Did controls with the same adversity exposures differ from depressed subjects in any of these metrics?

Response 8: In response to your specific inquiries regarding the mediation analysis, we provide the following comprehensive response:

  • Was the mediation analysis conducted only on the depressed patients? The mediation analysis was indeed conducted on both the depressed patients and the healthy controls. By including both groups in the analysis, we aimed to explore the relationships between the variables of interest in a comprehensive manner.
  • Were the same parameters examined for the healthy controls? Yes, we examined the same parameters for the healthy controls as we did for the depressed patients. We ensured that the measurements and variables used in the analysis were consistent across both groups. This approach allowed for a thorough and fair comparison between the groups and facilitated the identification of any potential differences or similarities in the variables under investigation.
  • According to the data,we collected the percentage of subjects with childhood trauma in the healthy groups was also low. Currently, the small sample size of our study data does not allow us  to answer this question. This is one of the limitations of our study. If conditions permit, we plan to address this limitation and explore the potential impact of EALs in a larger sample. This will provide a more comprehensive understanding.

Point 9: There are a very large number of non-standard abbreviations used throughout the paper, requiring a lot of back and forth to follow the text.  I would suggest a prominent box at the start of the manuscript that lists these abbreviations and can be referred back to by the reader.  If that is not possible, the full descriptions of the variables will need to be given in the Results and discussion sections.

Response 9: The list of abbreviations has been added according to your advice.

Point 10: Although the findings are certainly interesting, it was unclear how many associations were tested to identify the significant ones- could the authors clarify?

Response 10: To clarify the number of associations tested to identify the significant ones in the analysis, we would need to refer back to the specific analysis and determine the total number of associations examined. The provided analysis results mention three paths: paths a, b, and c, as well as an indirect path ab.

Based on the information provided, these four paths represent the associations tested in the analysis. Therefore, four associations were examined to identify the significant relationships between the variables. These associations include:

  • Association between ELAs and depressive symptoms (path c)
  • Association between childhood trauma exposure (ELAs) and LF-HRV (path a)
  • Association between LF-HRV and depressive symptoms (path b)
  • Indirect effect of childhood trauma exposure on depressive symptoms through LF-HRV (path ab)

It's important to note that in mediation analyses like the one described, multiple associations are typically examined to understand the complex relationships between variables. By assessing these associations, we can determine the direct and indirect effects and explore potential mediating mechanisms.

In our analysis, we conducted careful statistical tests on these associations to identify the significant ones. We used appropriate significance thresholds (e.g., p-values) and confidence intervals to establish the statistical significance of the relationships.

Point 11:The last line of the Conclusions section is part of the manuscript instructions and should be deleted.

Response 11: It has been deleted according to your advice.

Point 12: Some of the conclusions being drawn in the paper are based on very small numbers of subjects- this should be stated.

Response 12: Thank you for your advice,We have already highlighted the small sample size as a limitation in the article.

Point 13: Informed Consent statement: This does not appear to be correct- it repeats the statement on data availability.  Can the authors revise?

Response 13:Thank you greatly for your job,the following revisions were made.

Point 14:However, the approach is somewhat novel and the findings certainly interesting and clinically relevant. As the authors state, longitudinal studies are needed to confirm these observations on larger groups.

Response 14:Thank you for recognizing our work, this is just a preliminary study and we will subsequently increase the sample size as much as possible and conduct longitudinal studies.

Reviewer 3 Report

This study examined the impact of childhood traumatic experiences (EALs) on depression, cognitive function, and the heart-brain axis. The results revealed that EALs were associated with increased severity of depression symptoms, particularly emotional neglect. Significant differences in cognitive function were found between depressed patients and healthy controls, with deficits observed in global cognitive function and immediate memory. Reduced heart rate variability (HRV) was observed in depressed patients, and HRV partially mediated the association between childhood trauma exposure and depressive symptoms. The findings suggest a dysregulation of the autonomic nervous system in individuals with EALs. The study highlighted the overlap between autonomic regulation and emotional processing, emphasizing the role of HRV in emotion regulation. Despite some grammatical and typographical errors, the quality of writing is generally good, and the study's findings are appropriately summarized. The author needs to make minor language editing corrections.

This manuscript has some grammatical and typographical errors, but the quality of writing is generally good, and the study's findings are appropriately summarized. The author needs to make minor language editing corrections. Overall, this study is deemed suitable for publication in a journal.

Author Response

Response to Reviewer 3 Comments

Thank you very much for your recognition of our work. I have revised the overall grammar of the article based on your suggestions to enhance its readability. I greatly appreciate your valuable feedback.

Reviewer 4 Report

The manuscript highlight an interesting topic, Effects of Early Adverse Life Events on Depression and Cognitive Performance from the Perspective of Heart-Brain Axis,

Below are some comments to the authors

·        The importance and the rationale to conduct the study have to be explained better and in a smooth way for the readership

·        The sample size is important, the authors should explain how the sample size was reached at? Is there any sample size calculation done? How many patients and controls were initially approached? What was the response rate?

·        Sometimes, mental health is affected by the season, i.e “winter blues” , did the authors control this factor? Especially the design was cross-sectional

·        As I understood, the study was performed on patients from Asia (China), therefore, are the findings useful for this ethnicity? Or can they be used broadly?

Author Response

Response to Reviewer 4 Comments

Point 1: The importance and the rationale to conduct the study have to be explained better and in a smooth way for the readership.

Response 1: The target of this article is to clarify the impact of adverse childhood experiences on emotional and cognitive performance from the perspective of the heart-brain axis. This point highlighted the need for a better explanation of the importance and rationale behind conducting the study. We agree with this comment and have revised the introduction section to provide a more comprehensive and coherent explanation.

To ensure a smooth flow of information for the readership, we have restructured the introduction to provide a logical progression of ideas. We have also incorporated additional background information and contextual details to enhance the understanding of the study's rationale. By doing so, we believe that readers will have a clearer understanding of the importance of our research and its contribution to the existing body of knowledge. These revisions have been implemented in the revised manuscript, specifically in the introduction section. We believe that these changes have significantly improved the clarity and coherence of the manuscript, addressing the reviewer's comment.

Point 2: The sample size is important, the authors should explain how the sample size was reached at? Is there any sample size calculation done? How many patients and controls were initially approached? What was the response rate?

Response 2:

Sample size calculation:

We understand the importance of sample size calculation in ensuring the statistical power and validity of our study. But,There is no single formula for sample size calculation that applies universally to all situations and circumstances.We can only estimate a sample size based on the formula, the exact sample size still depends on the actual situation.We applied an equivalence test to calculate the sample size using the procedure “Tests for Two Mean in a 2×2 Cross-Over Design [Difference]” in PASS 11 software (NCSS, LLC. Kaysville, UT, USA). Five parameters were determined before calculating the sample size.

  • Power: set at 0.90
  • Alpha (significance level): set at 0.05, two-sided.
  • Diff0 (Mean Difference|H0): The difference between two means under the null hypothesis, which was set at 0.
  • Diff1 (Mean Difference|H1): The difference between two means under the alternative hypothesis.
  • S: standard deviation of paired difference.

Specifically, we first reviewed published literature. Then, we calculated the expected differences (Diff1) by linearly converting the literature-based differences. Last, we calculated the standard deviations of paired differences from the stand errors. Considering some practical issues (e.g. cost and time), the sample size was determined to be 45. Actually, the total sample comprised 97 participants, divided into two distinct groups: the Depression group (n = 48) and healthy controls (n = 49).

Number of patients and controls approached: Initially, we approached a total of 60 patients and 65 controls to participate in the study. These individuals were contacted following the inclusion/exclusion criteria specified in the Methods section of the manuscript. Finally, The total sample comprised 97 participants, divided into two distinct groups: the Depression group (n = 48) and healthy controls (n = 49). The details of the recruitment process, including the eligibility criteria, recruitment sources, and screening procedures, have been revised and provided in the revised manuscript.

Response rate: Out of the total number of individuals approached, 80% of depressed patients and 75% healthy controls agreed to participate in the study. The response rate was calculated by dividing the number of participants who accepted the invitation by the number of individuals initially approached. We believe that this response rate indicates a good level of interest and willingness of participants to contribute to the research.

Point 3: Sometimes, mental health is affected by the season, i.e “winter blues” , did the authors control this factor? Especially the design was cross-sectional.

Response 3: Previous clinical studies have certainly shown a higher prevalence of depression in the Winter , but the exact mechanisms are unknown , and seasonal influences were not the focus of this study. In addition, this study was conducted from march to June which is not during the winter season.Of course this is a limitation of this study, which was only a cross-sectional study due to time and cost considerations, and it would be a novel direction to sample in different seasons as you suggested.

Point 4: As I understood, the study was performed on patients from Asia (China), therefore, are the findings useful for this ethnicity? Or can they be used broadly?

Response 4: Previous studies have shown that heart rate variability and band power of EEG are altered in depressed patients in other ethnic groups.1-3Our study for an Asia population also comes to consistent conclusions. It suggests that alterations in the mind-brain axis are characteristic of the disease itself and are not related to ethnicity.

  1. Lee D, Kwon W, Heo J, Park JY. Associations between Heart Rate Variability and Brain Activity during a Working Memory Task: A Preliminary Electroencephalogram Study on Depression and Anxiety Disorder. Brain Sci. 2022;12(2):172. Published 2022 Jan 28. doi:10.3390/brainsci12020172

2.Čukić M, Savić D, Sidorova J. When Heart Beats Differently in Depression: Review of Nonlinear Heart Rate Variability Measures. JMIR Ment Health. 2023;10:e40342. Published 2023 Jan 17. doi:10.2196/40342

3.Hartmann R, Schmidt FM, Sander C, Hegerl U. Heart Rate Variability as Indicator of Clinical State in Depression. Front Psychiatry. 2019;9:735. Published 2019 Jan 17. doi:10.3389/fpsyt.2018.00735
